# Optimizing Computationally-Intensive Simulations Using a Biologically-Inspired Acquisition Function and a Fourier Neural Operator Surrogate

**John P. Lins[1] & Wei Liu[2]**
[1,2]Lawrence Livermore National Laboratory, [1]UC Santa Barbara
Livermore, CA 94551, USA
[1]johnlins@engineering.ucsb.edu, [2]liu56@llnl.gov

## Abstract

Computational modeling of physical phenomena has enabled researchers to acquire insight that was previously only observable from costly real-world experiments. The adoption of physics simulations has resulted in numerous advancements in fusion energy, seismic inversion/monitoring, and national defense. However, optimizing simulation studies requires many realizations of the intensive simulations. Manually tweaking control parameters and searching for optimal results can be tedious and inefficient. To tackle such obstacles in simulation studies, we found that differential evolution combined with the covariance matrix adaptation strategy could effectively optimize simulations while simultaneously behaving as an acquisition function to collect samples. The samples collected may be used to train intelligent surrogate models such as a Fourier neural operator (FNO). Once a surrogate is constructed, it could be used to accelerate the sampling of the optimization search space further. This methodology effectively optimized a hydrodynamic simulation modeled by systems of partial differential equations; it may also be extended to simulation optimization in other disciplines.

## 1 Introduction

**Idea**  Optimizing the control parameters of certain physics simulations needs to be approached as a black-box optimization problem when gradient information of the loss landscape cannot be computed from the simulation itself. To work around this issue, we designed a new genetic optimization algorithm in conjunction with constructing a differentiable surrogate model of the simulation using the samples that were already taken; our approach will enable us to efficiently optimize the simulation further and possibly perform gradient-based optimization.

**Surrogate**  This idea prompted us to examine the FNO's effectiveness at emulating the real simulation. Executing this task required us to take samples of the simulation. However, if we were to run a set of simulations with randomly or uniformly generated control parameters, then too many samples would crash; that is why we use genetic optimization first, to have a high quality data set.

**DECAF**  We propose DECAF (Differential Evolution [with] Covariance Adaptation [followed by optimizing a] Fourier [neural operator surrogate]), a novel solution. Our approach combines differential evolution with covariance matrix adaptation (Igel et al., 2007) to minimize undesirable artifacts while simultaneously behaving as an acquisition function to construct a data set of samples for the FNO surrogate. We then suggest ways to optimize the surrogate further to find a highly optimal set of parameters. We hypothesized that our approach would reduce the number of crashes through the course of sampling and create a high enough quality data set to train an FNO surrogate.

**Parameters**  Our control parameters were defined by the artificial coefficient of shear viscosity ($c_m$), bulk-viscosity dilatation ($c_b$), and thermal conductivity ($c_k$), as well as, ceiling values on artificial

bulk viscosity ($c_l$), thermal conductivity ($c_c$), and diffusivity ($c_f$). These artificial coefficients are used to enhance numerical stability and fidelity by smoothing shock waves interacting with materials.

Some choices of the artificial coefficients can render the simulation highly stiff, in which the time step size becomes drastically small. The minuscule time steps are often associated with non-physical numerical artifacts that can lead to instability in numerical simulations (Cook, 2007).

## 2 APPROACH

Differential evolution (DE) (Das & Suganthan, 2010) is a genetic optimization algorithm that optimizes non-convex loss landscapes. DE adheres to the following sequential process in Fig. 1.

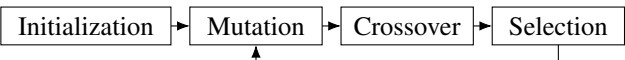

Figure 1: The sequential process of differential evolution

**Mutation**   Vanilla differential evolution (vanilla-DE) generates mutant vectors $\vec{m}_i \in \mathcal{M}$ in the following way. Here, $\alpha_m$ is the mutation factor and we set this value to $0.5$. $\mathcal{M}$ and $\mathcal{P}$ define the mutant and current populations respectively.

$$\vec{m}_i := \vec{v_1} + \alpha_m \cdot (\vec{v_2} - \vec{v_3}) \text{ where } \vec{v_1} \neq \vec{v_2} \neq \vec{v_3} \text{ and } \vec{v_1}, \vec{v_2}, \vec{v_3} \in \mathcal{P} \tag{1}$$

**DE-GM**   Unlike vanilla differential evolution, our approach of combining the covariance matrix adaptation strategy with differential evolution in this way uses a multivariate Gaussian to generate mutant vectors. We call our approach "Differential Evolution with Gaussian Mutation (DE-GM)."

$$X \sim \mathcal{N}(\vec{\mu}_\mathcal{P}, \vec{\sigma}_\mathcal{P}^2) \tag{2}$$

X is a random variable constructed using the mean $\vec{\mu}_\mathcal{P}$ and diagonal covariance entries $\vec{\sigma^2}_\mathcal{P}$ of the entire current population.

$$\vec{m}_i := X \tag{3}$$

This Gaussian is then sampled to construct a mutant population.

**Crossover**   Differential evolution uses crossover to retain some good genetic material from the previous generation. Crossover is performed by randomly swapping some elements of the mutant vectors with elements from the current population vectors to construct trial vectors $\in \mathcal{T}$. The trial vectors are later compared to the current population vectors during the selection process.

$$\mathcal{T}_{i,j} = \begin{cases} \mathcal{M}_{i,j} & rand(0,1) \geq \alpha_c \\ \mathcal{P}_{i,j} & otherwise \end{cases} \tag{4}$$

$\alpha_c$ is the crossover factor, we set this value to $0.2$.

**Selection**   After the test function or simulation has been evaluated, we iterate through the current and trial vectors with their corresponding losses. We select the next population ($\mathcal{N}$) in the following way.

$$\mathcal{N}_i = \begin{cases} \mathcal{T}_i & loss(\mathcal{T}_i) < loss(\mathcal{P}_i) \\ \mathcal{P}_i & otherwise \end{cases} \tag{5}$$

**Bounds**   We handled boundary conditions by replacing mutants that step outside the boundary with a random value within the defined boundary. For our simulation, each parameter was bound by its default value scaled up and down by a factor of 10.

**Convergence**   We eventually declared that the process has converged when the average loss of the population for the past $n$ iterations is less than the convergence constant $\alpha_v$. When optimizing the test function, we used $n = 10$, $\alpha_v = .001$, and when optimizing the actual simulation, we were more forgiving and used $n = 3$, $\alpha_v = 10$.

## 3 TESTING GENETIC ALGORITHMS

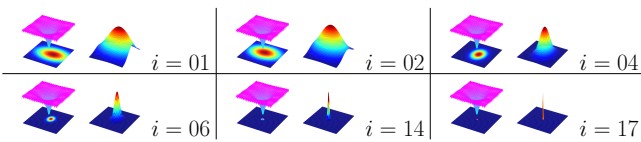

Figure 2: The multivariate Gaussian changing per iteration $i$

**Test Function**   We conducted experiments on a test function before optimizing the simulation itself. The Ackley function is ideal for testing because it has many non-optimal local extrema and one global minimum. We used a test function to make it computationally inexpensive to compare vanilla-DE with DE-GM. When we tested DE-GM on the test function, Fig. 2, intuitively, as the number of iterations $i$ increased $i \to \infty$, the population became less scattered $\vec{\sigma_{\mathcal{P}}}^2 \to \vec{0}$ while $\vec{\mu}_{\mathcal{P}} \to$ optimal minimum.

### 3.1 RESULTS AND COMPARISON

We collected data from 15 training sessions of both vanilla-DE and DE-GM. Independent t-test for difference of mean with an $\alpha$-value of 0.01 informed us that DE-GM converged 5.667 iterations earlier, with significance ($p = 0.003 \ll 0.01$), and at a 0.032 point lower final average loss ($-66.76\%$), without significance ($p = 0.512 \not< 0.01$). What mattered most was minimizing the number of iterations needed to converge, because fewer iterations means fewer computationally expensive samples need to be taken. For this reason, we decided to use DE-GM for optimizing the simulation.

DE-GM tends to exploit the loss landscape better, which is ideal for our task. Alternatively, vanilla-DE (and CMA-DE, DES, etc. (Arabas & Jagodzinski, 2020)) may cause mutation vectors to be flung far away from the rest of the population in the loss landscape if $\vec{v_2}$ and $\vec{v_3}$ are very different—wasting costly simulation samples.

## 4 DE-GM ON SIMULATION DATA

We experimented with a hydrodynamic simulation which modeled the behavior of a water droplet that underwent a hypersonic shock. This particular simulation was modeled by the Navier-Stokes momentum equation (Cook, 2009, eqn. 2), the advection-diffusion equation (Cook, 2009, eqn. 9), and the energy equation for multicomponent flows (Cook, 2009, eqn. 11).

**Objective**   The optimization objective was to minimize the number of time steps needed to reach a fixed point in time—fewer time steps due to a bigger step size tend to yield fewer undesirable non-physical artifacts. We defined this fixed point in time to be 0.2 $\mu s$. The loss landscape is discrete, and the input space is continuous. We also accounted for trials where the choice of control parameters caused the simulation to fail to reach the specified point in time; we declare this phenomenon a 'crash.' When this happened, we assigned the largest possible error to it: 9999999. By doing this, the next population would likely not incorporate parameters that cause crashes. Our population size was four, and we ran six training sessions.

### 4.1 RESULTS

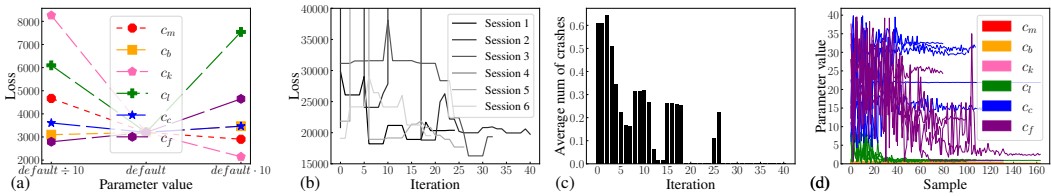

Figure 3: The optimization process over time provided valuable insights into the effectiveness of DE-GM: (a) sensitivity of parameters, (b) loss reduction over six consecutive sessions, (c) fewer crashes over iterations, (d) convergence of parameters.

Each parameter had been independently minimized and maximized while keeping the other five parameters constant at their default value, we observed that some are more sensitive to extreme values than others, Fig. 3(a).

Fig. 3(c) illustrates the average number of crashes that typically occurred in a given iteration step across all training sessions. Since the number of crashes decreased over each iteration, we reduced the amount of computational waste.

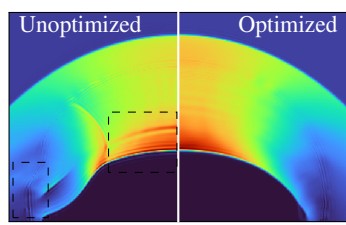

Fig. 3(d) illustrates where each parameter converged per training session, often at different values per session; this indicates that DE-GM did not find the same minimum for each session.

The pressure of the third training session is illustrated in Fig. 4. The first iteration ($loss = 31146$) showed some non-physical oscillations, potentially Gibbs oscillations (Gottlieb & Shu, 1997). The last iteration, now optimized, had much fewer time steps ($loss = 12793$) and showed minimal visual evidence of undesirable non-physical artifacts.

Figure 4: Simulated pressure field with randomly initialized v.s. optimized parameters

## 5 FOURIER NEURAL OPERATOR

One FNO layer update (Li et al., 2020; Kovachki et al., 2021) consists of passing input function $v(x)$ through both a local linear operator as well as a kernel integral operator; the two are then summed together and passed through a non-linear activation function.

$$v_{t+1}(x) := \sigma(W v_t(x) + \mathcal{F}^{-1}(\zeta[R_\theta \cdot (\mathcal{F}v_t(x))])) \tag{6}$$

$\mathcal{F}$ is the fast Fourier transform (FFT). $\sigma$ is a non-linear activation function—such as ReLU. $R_\theta$ is a transformation which contains the learnable parameters. $\zeta$ removes the higher Fourier modes. $W$ is the local linear operator.

We applied the standard 4-layer FNO architecture for our experiments. As a proof of concept, the neural operator only predicts the sixth snapshot given the fifth snapshot, Fig. 5. The FNO was trained in under 24 hours on a single node of an NVIDIA V100 GPU for 200 epochs with 488 training samples and 50 test samples.

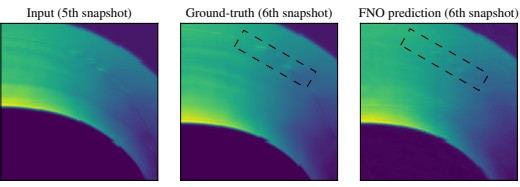

Not only can the trained FNO be evaluated efficiently, but it is also differentiable with respect to the control parameters—meaning we can further optimize using gradient descent.

Figure 5: FNO accurately predicting the pressure

By constructing a multilayer perceptron (MLP) to identify patterns indicating non-physical oscillations in the FNO output—which we know is possible according to the universal approximation theory (Hornik et al., 1989)—then we can use the FNO to explore the loss landscape.

## 6 SUMMARY

DE-GM can effectively explore simulation settings while minimizing the number of heavily penalized samples and conserving computational resources. Our study found that an accurate FNO surrogate can be constructed from sparse samples in time evolution from the simulation taken with the DE-GM acquisition function. The approach studied here may also be used to optimize other types of numerical experiments in which users can flexibly define optimization objectives.

**Limitations and Broader Impacts** There are no apparent negative societal impacts to this work. A limitation of our study is that training an MLP to approximate the loss from visual data was untested.

ACKNOWLEDGMENTS

We would like to thank Dr. Britton Olson for his assistance related to LLNL's MIRANDA software, Dr. Andrew Cook & Calvin Young for their guidance with the hydrodynamic simulation, and Dr. Qingkai Kong for his assistance regarding the Fourier neural operator. This work was performed under the auspices of the U.S. Department of Energy by Lawrence Livermore National Laboratory under contract DE-AC52-07NA27344. Lawrence Livermore National Security, LLC.

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
