# OpenReview forum: "Optimizing Computationally-Intensive Simulations Using a Biologically-Inspired Acquisition Function and a Fourier Neural Operator Surrogate"
_ICLR.cc/2024/Workshop/AI4DiffEqtnsInSci — AI4DiffEqtnsInSci @ ICLR 2024 Poster_

### Official Review · Reviewer_WXQP · 2024-02-25
**Promising idea with unclear motivation and limited evaluation**

**Rating:** 4
**Confidence:** 3

**Review:**

**Summary**: The papers propose a method to optimize parameters of physics simulations using samples generated from a genetic optimization algorithm, which is then used to construct an FNO surrogate. The authors evaluate the performance of the FNO and propose further optimization that could be applied to this surrogate via gradient-based algorithms.


**Strengths**
1. Combining differential evolution as a precursor to surrogate models seems like an exciting direction. The authors evaluate their results on computationally intensive tasks such as hydrodynamic simulations.

**Areas of improvement**:

1. The idea of combining multiple techniques lacks clarity and motivation. The choice of CMA-DE and FNO does not seem convincing, as the manuscript lacks details about clear advantages compared with the existing state-of-the-art.
2. There is no evaluation of different sampling strategies or other optimization methods for surrogates, which may have provided better insights into their algorithm choices. There are no details about computational performance analysis.
3. For the CMA-DE and FNO - The CMA-DE method has previously been studied in the literature, but this work doesn't cite any previous publications and claims the technique as a novel [arabas2019, grady2023]. The role of FNO in the work is unclear, and the experiment details are missing.

[arabas2019] Arabas, Jarosław, and Dariusz Jagodziński. "Toward a matrix-free covariance matrix adaptation evolution strategy." IEEE Transactions on Evolutionary Computation 24.1 (2019): 84-98.

[grady2023] Grady, Thomas J., et al. "Model-parallel Fourier neural operators as learned surrogates for large-scale parametric PDEs." Computers & Geosciences (2023): 105402.

---

### Official Review · Reviewer_mnSx · 2024-02-26
**The proposed approach is potentially of interest for calibrating and tuning numerical simulations across different research fields.**

**Rating:** 7
**Confidence:** 3

**Review:**

The authors propose a methodology using differential evolution combined with covariance matrix adaptation to optimize physical numerical simulations efficiently and then employ the samples collected to train a surrogate model, namely fourier neural operator (FNO). This surrogate may be used to enhance the optimization process by accelerating sampling. Although this approach was successfully applied to optimize a hydrodynamic numerical simulation, it has potential for application in other disciplines requiring physical simulation optimization.

The present study is potentially of interest for calibrating and tuning numerical simulations across different research fields, such as climate science. While the aim of the paper is clearly defined in the manuscript, the text would benefit from further specific details to better understand the results and their implications. The theoretical concepts are concisely presented. The figures and the experiments are in general sound, but the text can better highlight key points (Section 4.1).
Specific comments:
- Section 2 – Approach. The mutation factor (αm) and the crossover factor (αc) seem to be arbitrarily chosen to 0.5 and 0.2, respectively. A motivation or strategy defining these factors is missing. Moreover, how sensitive are the results presented here to these factors.
- Section 2 – Approach. How is the loss defined (selection)?
- Section 4 – CMA-DE on Simulation data. Why is the fixed point in time defined as 0.2 µs?  How do the population size and training sessions affect the results?
- Section 5 – FNO. Non-linear activation function “such as ReLU”, is this the function used? “Trained overnight” can be better defined, it is a bit vague.
- Limitations. To me this approach requires to a good extent expert knowledge (e.g., defining control parameters and their likely ranges, or “crash”).

---

### Meta-Review · Area_Chair_kxvQ · 2024-02-28

**Recommendation:** Accept (Poster)

**Metareview:**

Dear Authors,

Thank you for submitting the draft.

Both reviewers agree that the proposed methodology is potentially interesting. However, they do also raise some major points of concern. It is expected that authors will be addressing comments by the reviewers in the final draft.

regards

AC

---

### Decision · Program_Chairs · 2024-02-29

Accept (Poster)